# Effect of 24-Week, Late-Evening Ingestion of a Calcium-Fortified, Milk-Based Protein Matrix on Biomarkers of Bone Metabolism and Site-Specific Bone Mineral Density in Postmenopausal Women with Osteopenia

**DOI:** 10.3390/nu14173486

**Published:** 2022-08-24

**Authors:** Catherine Norton, Manjula Hettiarachchi, Rachel Cooke, Marta Kozior, Hilkka Kontro, Rosemary Daniel, Philip Jakeman

**Affiliations:** Health Research Institute and Department of Physical Education and Sport Sciences, University of Limerick, V94 T9PX Limerick, Ireland

**Keywords:** randomized control trial, milk protein matrix, nutrient intervention, nutrient timing, bone remodelling, bone turnover markers, bone health, osteopenia, postmenopausal women

## Abstract

Dietary calcium intake is a modifiable, lifestyle factor that can affect bone health and the risk of fracture. The diurnal rhythm of bone remodelling suggests nocturnal dietary intervention to be most effective. This study investigated the effect of daily, bed-time ingestion of a calcium-fortified, milk-derived protein matrix (MBPM) or control (CON), for 24 weeks, on serum biomarkers of bone resorption (C-terminal telopeptide of type I collagen, CTX) and formation (serum pro-collagen type 1 N-terminal propeptide, P1NP), and site-specific aerial bone mineral density (BMD), trabecular bone score (TBS), in postmenopausal women with osteopenia. The MBPM supplement increased mean daily energy, protein, and calcium intake, by 11, 30, and 107%, respectively. 24-week supplementation with MBPM decreased CTX by 23%, from 0.547 (0.107) to 0.416 (0.087) ng/mL (*p* < 0.001) and P1NP by 17%, from 60.6 (9.1) to 49.7 (7.2) μg/L (*p* < 0.001). Compared to CON, MBPM induced a significantly greater reduction in serum CTX (mean (CI_95%_); −9 (8.6) vs. −23 (8.5)%, *p* = 0.025 but not P1NP −19 (8.8) vs. −17 (5.2)%, *p* = 0.802). No significant change in TBS, AP spine or dual femur aerial BMD was observed for CON or MBPM. This study demonstrates the potential benefit of bed-time ingestion of a calcium-fortified, milk-based protein matrix on homeostatic bone remodelling but no resultant treatment effect on site-specific BMD in postmenopausal women with osteopenia.

## 1. Introduction

Dietary calcium intake is a modifiable, lifestyle factor that can affect bone health and the risk of fracture. Based on studies of calcium balance, a recommended daily intake (RDI) of 1500 mg/d achieves neutral calcium balance in postmenopausal women [1]. However, reported habitual dietary intake of calcium often falls below the recommended level for calcium balance [2] invoking a range of nutritional strategies to increase calcium intake by dietary or supplemental means [3]. Meta-analyses [4,5,6] indicate calcium supplementation in postmenopausal and elderly women is associated with a lesser decrease in bone mineral density (BMD), particularly in the early years of supplementation, and a greater effect in those with low reported dietary calcium intake. In contrast, cross-sectional and longitudinal observational studies concluded that increasing calcium intake, commonly achieved by calcium supplements, did not provide any benefit for hip or lumbar BMD [7], and that post-menopausal bone loss is unrelated to habitual dietary calcium intake in postmenopausal women with osteopenia [8].

Bone remodelling exhibits a unimodal diurnal rhythm with a nocturnal peak and daytime nadir [9,10], and therefore timing the ingestion of supplementation to coincide with the nocturnal peak of bone remodelling is an optimal, novel strategy for many in this vulnerable population. Additionally, superior to a calcium only supplement, the matrix of nutrients contained within dairy-based food products contain potent co-modulators of bone remodelling [11] that are pluripotent in effect when consumed as a matrix [12]. We have recently advanced a strategy that optimises the bio-efficacy of supplementary calcium through incorporation within a milk based protein matrix (MBPM), and optimal timing of MBPM intake to the diurnal pattern of bone remodelling in postmenopausal women with osteopenia [13]. Briefly, this study demonstrated the acute effects of bed-time ingestion of a calcium-fortified MBPM on bone remodelling in a block-randomized, within-subject crossover design resulting in a significant mean reduction (−32%) in the biomarker of bone resorption, serum C-terminal cross-linked telopeptide of type I collagen (CTX), and a mean increase (+24%) in the biomarker of bone formation, procollagen type 1 amino-terminal propeptide (P1NP), at 0–4 h post-ingestion. In addition, a significant mean reduction (−10%) in 24 h urinary biomarker of bone resorption, N-telopeptide cross-links of type I collagen (NTX), was observed [13]. Our aim in this current study was to investigate the benefits observed in the acute setting longitudinally and report the outcome of a 24 week, randomised, control trial investigating the effect of bed-time ingestion of ingestion of MBPM, fortified with calcium and vitamin D on biomarkers of bone metabolism and site-specific bone mineral density in postmenopausal women with osteopenia. An isoenergetic, maltodextrin acted as a control (CON).

## 2. Materials and Methods

The University of Limerick, Faculty of Education and Health Sciences Research Ethics Committee (2018_04_05_EHS) approved the study, which was conducted in accordance with the ethical standards outlined in the most recent version of the Declaration of Helsinki and registered with clinicaltrials.gov identifier NCT03701113. Participants were informed of the purpose of the study and all known risks before providing written, informed consent. Figure 1 presents a flow diagram of the participant recruitment, enrolment, completion and analysis. The projected number of participants (*n* = 30 per group) required in the project proposal was successfully achieved with an average compliance of ~80%.

### 2.1. Study Design

A block-randomised, controlled study was conducted among healthy, postmenopausal women with osteopenia receiving, a daily milk-based protein supplement (MBPS) or isoenergetic maltodextrin control (CON) at bed-time for a period of 24 weeks. Principal outcome measures were the change from baseline at 24 weeks in biomarkers of bone resorption (C-terminal telopeptide of type I collagen, CTX, ng/mL) and formation (pro-collagen type 1 N-terminal propeptide, P1NP, ng/mL) in fasted blood, and site-specific aerial bone mineral density (BMD) and AP spine trabecular bone score (TBS). Based on previous studies of a 24 week intervention [14] a sample size of 30 subjects per treatment group attains a statistical power of >80%, and alpha of 0.05 for a measurable change in BMD of 0.017 g∙cm^−2^ for MBPM compared to CON.

### 2.2. Participant Recruitment and Progression

A convenience sample of 151 postmenopausal women aged 50 to 70 years old were recruited from the University of Limerick Body Composition (ULBC) Study database by invitation. ULBC subjects had preassigned alphanumeric ID codes that continued to the present study. Subject volunteers were free-living, lactose tolerant, and devoid of treatment for osteoporosis or drugs affecting calcium absorption. Dual energy X-ray measurement of AP spine (L1–L4) BMD and TBS was conducted and analysed in accordance with the International Society for Clinical Densitometry [15] and confirmed eligibility (AP spine (L1–L4) BMD T scores < −1.0, ≥−2.5). Ninety-seven volunteers confirmed as osteopenic submitted to clinical examination and, blood screen and provided a 7-day weighed food dietary intake record. Following screening four volunteers declined to participate further and two were excluded on low (>−2SD) reported habitual calcium intake. The remaining 83 volunteers progressed to the RCT and were block randomized to receive either a milk-based protein supplement (MBPM) or an isoenergetic, maltodextrin control (CON) for 24 consecutive weeks (Figure 1). 

### 2.3. Mean Daily Intake (MDI) and Supplement Intake

Following instruction from a registered dietitian, participants recorded dietary intake for seven consecutive days (5 weekdays, 2 weekend days) by weighed food and fluid log. Participants reported food, quantity of leftovers, brand name, quantity, cooking method, type, time, and location of meal preparation. Subsequent coding and analysis was completed using Nutritics^™^ Dietary Analysis Software (Nutritics^™^ software version 4.0 for Ireland). Reported intakes of energy, macronutrients, calcium and Vitamin D was compared to reference intakes (DRIs) developed by the Food and Nutrition Board (FNB) at the Institute of Medicine (IOM) of The National Academies [16].

Participants received a supplemental intake of MBPM or CON packed in individual packets to be consumed prior to bed-time. The nutrient composition of MBPM and CON comprised an additional 0.3 g/kg body mass protein and 0.3 g/kg body mass carbohydrate in the MBPM group and an equivalent amount of carbohydrate energy, i.e., 0.6 g/kg body mass in the CON arm of the study. Appendix A report the prescribed dietary supplement intakes of CON and MBPM, respectively. Compliance to supplement use was informed by countback of packets.

### 2.4. Anthropometry and Bone Densitometry

To standardize test conditions and tissue hydration, participants were instructed to refrain from strenuous exercise in the 24-hour period before attending and to present after an overnight fast. Participants consumed 500 mL water 1 h before, and were instructed to void and defecate, if required, immediately prior to measurement. Height was measured to the nearest 0.1 cm by using a stadiometer (Seca) and BM was measured to the nearest 0.1 kg (MC-180MA; Tanita UK Ltd., Hayes, UK). Bone densitometric and whole-body compositional analysis was performed using DXA (Lunar iDXA™ with enCORE™ v.18 with integrated TBS software, GE Healthcare, Chalfont St Giles, Bucks, UK) (Lunar iDXA™ with enCORE™ v.18 with integrated TBS software and analysed in accordance with the International Society for Clinical Densitometry [15]. Calibration by means of a phantom spine was performed daily. 

### 2.5. Clinical Biochemistry

Blood samples were obtained by venipuncture of an antecubital vein. Serum and plasma were separated by centrifugation at 10,000× *g* at 4 °C for 5 min and aliquots frozen at −80 °C before batch analysis. Serum CTX, P1NP and total 25-hydroxyvitamin D were analysed by 2-site immunometric assay using electrochemiluminescent detection (Roche Cobas e411, Roche Diagnostics, UK. The interassay CV was 5.3% for CTX, 4.5% for P1NP and <8% for 25(OH) Vit D. 

### 2.6. Statistical Analysis

Shapiro–Wilk test was applied to check for normality of the data that are presented as the mean ± SD unless stated otherwise. The treatment effect was calculated as the change in biomarkers of bone turnover and site-specific DXA-derived bone mineral density from baseline to 24 weeks. These data were found to be normally distributed and analyzed by univariate ANOVA with treatment (CON compared with MBPM) as fixed factors. Statistical analysis was performed by using PASW Statistics 20.0 for Windows (IBM, Chicago, IL, USA). Significance (2-tailed) was set at *p* < 0.05 for all analyses.

## 3. Results

### 3.1. Subject Characteristics

The baseline characteristics of the subject population are representative of postmenopausal women aged between 51 and 75 years old, height 148 to 175 cm, weight 44 to 113 kg and body mass index (BMI) 18 to 41 kg∙m^−2^ (Table 1). Subjects were confirmed as osteopenic with BMD T-score of the L1-L4 region of the spine. There were no statistically significant differences in the baseline characteristics between CON and MBPM.

### 3.2. Habitual and Intervention Dietary Intake

Fifty-five subjects returned a viable 7-day weighed food and fluid log. Reported, habitual mean daily intake, demonstrated adequacy among 85% of the study population for dietary protein (1.1 ± 0.3 g·kg^−1^·d^−1^) when assessed relative to IOM RDA (0.8 g·kg^−1^·d^−1^). Absolute habitual daily protein intake was 74 ± 16 g·d^−1^. The mean reported habitual daily calcium intake (954 ± 488 mg·d^−1^) surpassed the lowest acceptable intake (800 mg·d^−1^) for 51% of participants with only 24% of the population reporting calcium intakes at or above the IOM RDA of 1200 mg·d^−1^. On average, habitual, daily vitamin D intake was 13 ± 17 µg·d^−1^, and twenty percent of the study population met the current IOM RDA for vitamin D (15 µg d^−1^). Reported Vit D intakes are similar to those of other nationally representative studies, and the range is high between subjects. Supplementation was the major contributor to overall vitamin D intake. Those with higher reported Vit D intakes reported use of Vit D supplements. Table 2 presents habitual daily energy, macronutrient, calcium, and Vitamin D intake for MBPM and CON groups at baseline. Figure 2 depicts 7-day habitual protein, calcium, and vitamin D distribution in 4 h epochs over 24 h for MBPM and CON.

Appendix A provide an individual standardized nutrient intake of the dietary supplement for CON and MBPM, respectively. Compared to habitual intake, the addition of the supplement to CON and MBPM augmented subjects’ habitual daily total energy intake by 11% from 1552 ± 277 (23.6 ± 6.4 kcal·kg^−1^·d^−1^) to 1717 ± 270 (26 ± 6.4 kcal·kg^−1^·d^−1^) kcal·d^−1^ (*p* < 0.01). In CON, the increase in total energy resulted from a 27% increase in carbohydrate intake from 2.437 ± 0.7 to 2.94 ± 0.7 g·kg^−1^·d^−1^ (*p* < 0.01). For MBPM, the equivalent energy was supplied by a 30% increase in protein intake to 1.5 ± 0.3 g·kg^−1^·d^−1^ (*p* < 0.01). In addition, the average daily calcium for MBPS increased by 107% from 957 ± 514 to 1821 ± 538 mg·d^−1^ (*p* < 0.01), and vitamin D by 20% from 11.6 ± 15 to 12.5 ± 15 µg·d^−1^ (*p* < 0.01). Figure 2 presents the projected habitual + supplemental nutrient intake divided into 4 h epochs for each group indicating the temporal difference in nutrient intake between the treatment groups over the 24-week intervention.

### 3.3. Vitamin D Status

Serum 25(OH)vitamin D ranged from 40.0 to 113.0 nmol/L. Only 3 of the 65 participants recorded a 25(OH) vitamin D below 50 nmol/L. No difference in 25(OH)D was observed between CON (72.9 ± 16.0) and MBPS (70.3 ± 14.8) at baseline (*p* = 0.490). 

### 3.4. Change in Biomarkers of Bone Turnover (BTM)

The absolute and relative, percentage, change in serum biomarkers of bone resorption, CTX, and formation, P1NP, is presented in Figure 3. Serum CTX decreased by 9%, from 0.394 (0.073) to 0.316 (0.052) ng/mL (*p* < 0.001) in CON and 23%, from 0.547 (0.107) to 0.416 (0.087) ng/mL (*p* < 0.001) in MBPM. A statistically significant greater absolute (*p* = 0.047) and percentage (*p* = 0.025) decrease in CTX following intervention was observed in MBPM compared to CON (Figure 3a,b). Serum P1NP decreased by 19%, from 51.8 (8.6) to 40.2 (7.0) μg/L (*p* < 0.001) in CON and by 17%, from 60.6 (9.1) to 49.7 (7.2) μg/L (*p* < 0.001) in MBPM. Neither absolute (*p* = 0.592) or percentage (*p* = 0.802) change in P1NP between CON and MBPM following intervention was statistically significant (Figure 3c,d).

### 3.5. Bone Mineral Density (BMD) and Trabecular Bone Score (TBS)

The absolute and relative, percentage, change in site specific BMD and TBS is presented in Table 3. No significant effect of treatment with CON or MBPM was observed for site specific BMD of the AP spine (*p* = 0.427), dual femoral neck (*p* = 0.478) or trabecular bone score for the AP spine (*p* = 0.619). 

## 4. Discussion

This study was designed to investigate the effect of a 24-week, daily, nutrient supplement in support of bone health in postmenopausal women with osteopenia. The rationale for the study was provided by the outcome of a previous study of the acute (4 h) effect of bed-time ingestion of a calcium and vitamin D fortified supplement that induced a −32% and +24% mean change in the diurnal pattern of biomarkers of bone resorption (CTX) and formation (P1NP), respectively, and the putative mechanisms by which the MBPM induced an acute change in bone remodelling activity [13]. For this study we adopted a RCT design in which health related outcome related to bone metabolism were measured by change in CTX and P1NP to inform the potential long-term change in homeostatic rate of bone remodelling, coupled with the measurement of change in site-specific BMD and trabecular bone score as a clinical reference. 

Dietary intake of the present cohort of Irish women with osteopenia was compared to the Irish National Adult Nutrition Survey (NANS) [16]. We found the mean total energy (kcal) consumed per day to be similar to the mean energy intake in the NANS study (1552 ± 382 kcal). Similarly, the macronutrient contribution to total energy, (45.3 ± 7 vs. 46 ± 5.9% carbohydrate, 19.7 ± 3.3 vs. 18.3 ± 2.8% protein, 36.3 ± 6 vs. 35.5 ± 6.9% fat) and calcium intake (954 ± 488 vs. 995 ± 573 mg/day) was comparable to NANS. Reported Vit D intakes were similar to those of other nationally representative studies [17]. Supplementation with Vit D was the major contributor to overall vitamin D intake. The prevalence of multivitamin/single nutrient use is high in this population generally. Among women not consuming a vitamin D-containing supplement, mean daily intakes were approximately half the current recommendation and align with Irish national dietary data [17]. To these habitual diets was added a daily, non-habitual, bed-time, supplemental with an average energy value of 166 kcal. The nutrient value ascribed to the MBPM supplement was 22.2 g protein, and 875 mg calcium which increased the mean daily, and specifically, bed-time, energy, protein, and calcium intake, by 11, 30, and 107%, respectively. A matched energy intake in CON provided a 30% increase in carbohydrate intake only. Compliance to supplement intake, assessed by countback of supplement packets, was greater than 80% in both groups. 

The principal outcome of feeding a calcium and vitamin D fortified milk-based protein, timed to coincide with the diurnal peak rate of bone remodelling in postmenopausal women with osteopenia was to affect a statistically significant change in serum biomarkers of bone remodelling. Measured by the International Osteoporosis Foundation’s (IOF) endorsed biomarkers of bone resorption (CTX) and formation (P1NP) [18], the rate of bone remodelling is considered a risk factor of fracture that is independent of BMD, and high levels of remodelling may predict fracture risk in postmenopausal women [19]. CTX and P1NP are by-products of type I collagen turnover and, as protein constitutes ~50% of the volume of bone and approximately one-third of its mass, the turnover rate of type I collagen is important to the maintenance of bone structure and bone health. However, bone remodelling is subject to diurnal rhythm and acutely modified by nutrient ingestion and physical activity [9,20]. Indeed, previous [21] and recent [22] stable isotope studies of collagen metabolism indicate collagen turnover to be acutely sensitive to nutritional modulation. These extraneous factors influence the measurement of serum biomarkers of bone metabolism making it difficult to assess the change in homeostatic remodelling attributable to an intervention. In this study, the influence of prior feeding and/or physical activity was negated by sampling following 24h restricted physical activity and an overnight (10 h) fast. The influence of diurnal rhythm was also negated by sampling at the same time pre and post the 24-week intervention. It is probable, therefore, that the observed change in serum CTX and P1NP is representative of an underlying change in homeostatic bone remodelling activity. The magnitude of the effect of MBPM on bone remodelling activity was to reduce CTX and P1NP by an average of 23% (−0.131 ng/mL) and 19% (−49.7 μg/L), respectively. It is encouraging to note that this magnitude of effect resulting from a long-term, nutrient-based, intervention is equivalent to 41 and 50% of the effective change in CTX and P1NP, respectively, achieved by pharmacological intervention with oral bone resorption inhibitors (bisphosphonates) [23]. 

Equally, there is a strong link between dairy-based calcium supplementation and bone remodelling [24], linked to the high bioavailability and absorption of calcium, and reduced secretion of parathyroid hormone, a key regulator of bone remodelling activity [25]. It follows that calcium intake induces a transient remodelling and markers of bone metabolism react accordingly. To this effect it is reported that elevated parathyroid function and high bone remodelling typically found in elderly women can be normalized by a substantive increase in habitual calcium intake (~3 g/d) [26]. A key factor influencing the bio-efficacy of exogenous calcium intake is calcium uptake in the intestine. Calcium is absorbed from food in the intestine by active and passive transport. Passive absorption of calcium accounts for only 10–15% of calcium intake, hence there is reliance on the active transcellular pathway of intestinal calcium absorption in which the transfer of luminal calcium across the apical brush-border calcium channels is extruded across the basolateral membrane carrier protein, calbindin. Upregulation of calbindin in the intestine is promoted by an increase in calcitriol (1.25(OH)_2_D), the active form of vitamin D, in response to low circulating calcium, regulated by parathyroid hormone. The regulation of calbindin protein is vitamin D dependent, with optimal absorption of endogenous calcium plateauing at a serum [25(OH)D] in the region of 80 nmol/L [27]. The average [25(OH)D] for participants in this study was >70 nmol/L, with only three subjects recording <50 nmol/L, thus vitamin D related compromise to active absorption of the supplementary calcium intake should not have been evident in this subject cohort.

The present RCT found no significant change in site-specific bone mineral density or trabecular bone score resulting from the 24-week period of supplementation with MBPM. This outcome is consistent with other RCTs of supplemental calcium that report no effect of increased calcium intake on the rate of bone loss [28]. Indeed, there is considerable debate as to whether the typical dietary range of calcium intake, or supplemental calcium intake, influences the preservation or therapeutic change in bone mineral density in postmenopausal women. Following menopause, it is the reduction in oestrogen that is the primary cause of loss of bone mass, resulting in an inability of osteoblasts to maintain rates of bone formation equivalent to those of resorption [29]. Supplemental calcium intake may assist in buffering the increased calcium loss, but the deposition of calcium would be dependent on the restoration, and possible increase, in osteoblastic activity. In this respect, the high bone remodelling rate that is characteristic of postmenopausal osteoporosis, uncoupled to favour net resorption, exacerbates the rate of bone loss. A reduction in homeostatic remodelling rate, as observed in this study, may serve to reduce the rate of bone loss but, in the absence of a stimulus to osteoblastic activity may not be sufficient to restore bone mass. Supplemental calcium may, therefore, be surplus to requirement for bone tissue, leading to calcium excretion, or deposition in extraskeletal tissue. The extant evidence from long-term RCTs of calcium supplementation in support this view [5,6]. These studies report only a modest increase in BMD that is independent of habitual calcium intake [26], and evident primarily during the first 12 months [28], do not progress over time, and can be explained by a reduction in the bone remodelling space and skeletal space for calcium deposition [8]. 

Bone metabolism exhibits potent circadian rhythms that are modifiable by dietary intake. Notable in the outcome of our previous [13] study is the strong bidirectional and temporal relationship between calcium levels and serum PTH that contribute to the rhythmic activity in bone resorption. We considered the non-habitual addition of bed-time calcium and suppression of the peak night-time levels of PTH which, if repeated chronically over time (24 weeks), would result in downregulation of the bone remodelling rate, as demonstrated by the lower post-intervention serum CTX. A chronotherapy approach is supported by recent studies of the acute effect of timing of drug administration (the recombinant PTH, teriparatide™) on the diurnal rhythm of biomarkers of bone remodelling [30]. In a similar approach to that adopted in the current study, a 12-month follow-up intervention optimizing the timing of drug intake was found to affect a greater increase in bone mineral density in postmenopausal, osteoporotic women [31]. On the basis of these studies, we advocate the coordination of the timing of dietary intervention to the diurnal rate of bone remodelling as an effective strategy to improve the therapeutic efficacy of nutrient interventions in postmenopausal women with osteopenia.

## 5. Conclusions

A 24-week RCT investigating the long-term effects daily, bed-time ingestion of a milk-based, protein supplement fortified with calcium and vitamin D, in postmenopausal women with osteopenia resulted in a statistically significant reduction in biomarkers of bone remodelling, but no change in site-specific bone mineral density or trabecular bone score. Of note, the magnitude of this effect approximates to 50% of that achieved by pharmacological intervention over a similar time period. We conclude that a late-evening supplement of calcium-fortified milk protein effects a beneficial decrease in the homeostatic rate of bone remodelling in persons at risk of degenerative bone disease and recommend that future experimental protocols, investigating the effect of nutrient supplementation on bone metabolism and bone health, should consider a chronotherapeutic approach to their design and implementation.

## Figures and Tables

**Figure 1 nutrients-14-03486-f001:**
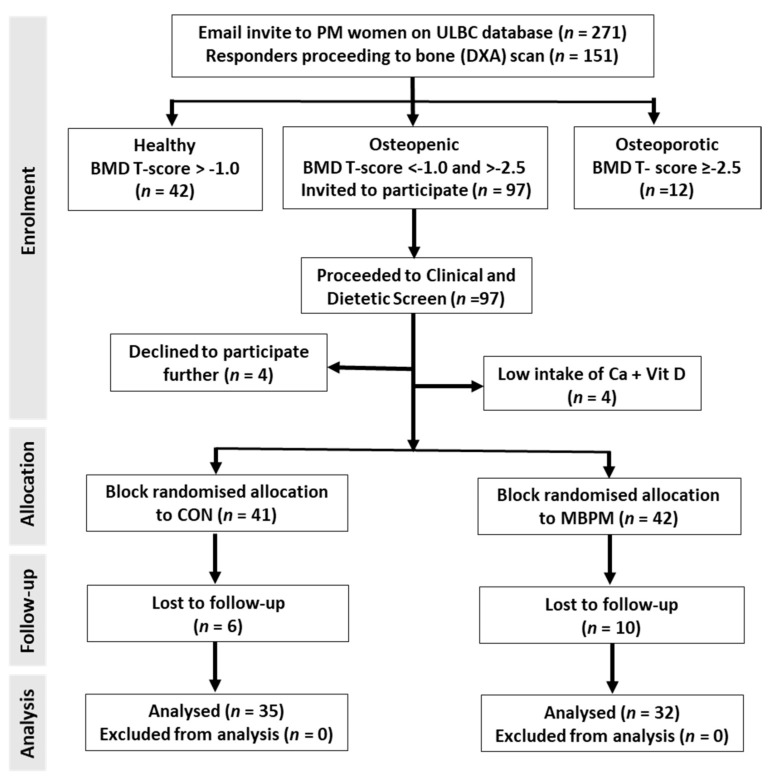
CONSORT flow diagram of participant enrolment, allocation, follow-up, and analysis.

**Figure 2 nutrients-14-03486-f002:**
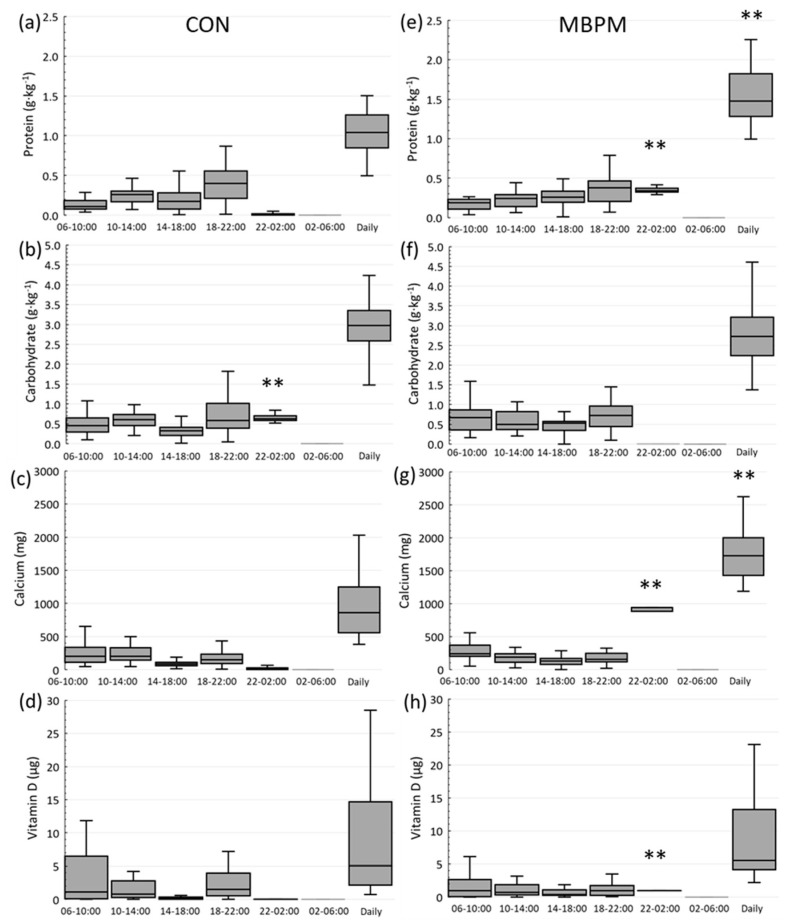
Four-hour epoch distribution of habitual nutrient intake and supplements for CON (**a**–**d**) and MBPM (**e**–**h**). ** denotes significant difference between CON and MBPM, *p* < 0.05.

**Figure 3 nutrients-14-03486-f003:**
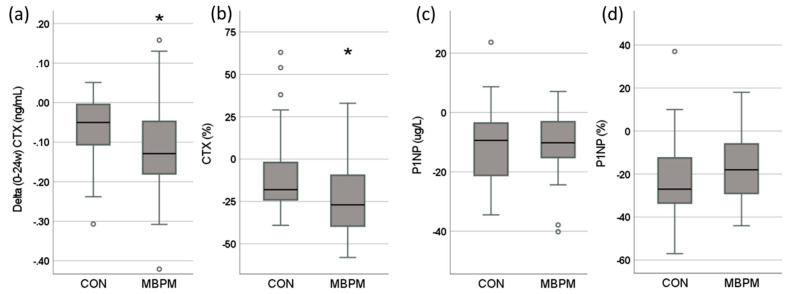
Absolute and relative change, from 0 to 24 weeks, in the homeostatic level of a biomarker of bone resorption (CTX; (**a**,**b**)) and bone formation (P1NP: (**c**,**d**)) following 24-week ingestion of CON and MBPM. * Denotes significant difference between CON and MBPM, *p* < 0.05.

**Table 1 nutrients-14-03486-t001:** Subject population (*n* = 67).

		ALL (*n* = 67)			CON (*n* = 35)			MBPM (*n* = 32)		
Characteristic	Unit	Mean	SD	Range	Mean	SD	Range	Mean	SD	Range
Age	y	62.8	6.0	51 to 74	63.6	6.3	51 to 75	61.9	5.8	51 to 74
Height	cm	161.0	5.2	148 to 175	160.7	4.2	151 to 171	161.3	6.1	148 to 175
Weight	kg	67.8	11.8	44 to 113	70.2	12.7	54 to 113	65.3	10.4	44 to 96
BMI	kg∙m^−2^	26.2	4.3	18 to 41	27.2	4.7	20 to 41	25.1	3.6	18 to 34
AP Spine (L1–L4) BMD	T-score	−1.5	0.4	−2.4 to −1.0	−2.3	0.4	−2.3 to −1.0	−1.5	0.4	−2.4 to −1.0
Trabecular Bone Score	T-Score	−1.7	0.9	−3.7 to 0.1	−1.5	0.8	−3.1 to 0.1	−2.0	0.9	−3.7 to −0.5

**Table 2 nutrients-14-03486-t002:** Baseline 7-day averaged daily nutrient intake for the calcium-fortified, milk-derived protein matrix (MBPM) and maltodextrin (CON) groups (*n* = 55).

	Average Habitual Daily Intake in CON Group						
	Energy (kcal·kg^−1^)	Protein (g·kg^−1^)	Carbohydrate (g·kg^−1^)	Fat (g·kg^−1^)	Protein (g)	Calcium (mg)	Vitamin D (ug)
Mean	22	1.0	2.4	0.8	73	951	13.3
SD	5	0.3	0.7	0.3	14	473	18.2
Min	10	0.5	1	0.3	43	384	1.0
Max	34	1.5	4.3	1.5	110	2032	80.0
	**Average Habitual Daily Intake in MBPM Group**						
Mean	25	1.2	2.8	0.9	75	957	11.6
SD	7	0.4	0.8	0.3	19	514	15.0
Min	12	0.7	1.4	0.3	53	505	1.0
Max	39	1.9	4.6	1.6	122	2805	62.0

Note. There were no statistically significant differences between groups (*p* > 0.05).

**Table 3 nutrients-14-03486-t003:** Absolute and relative percentage change in bone mineral density (BMD) and trabecular bone score (TBS) following 24-week intervention in CON and MBPM.

		AP Spine BMD (g∙cm^−2^)				Dual Femur BMD (g∙cm^−2^)				TBS (g∙cm^−2^)			
		Week		∆_0–24_		Week		∆_0–24_		Week		∆_0–24_	
		0	24	g∙cm^−2^	%	0	24	g∙cm^−2^	%	0	24	g∙cm^−2^	%
CON	mean	1.016	1.017	0.001	0.10	0.910	0.904	−0.004	−0.47	1.339	1.329	−0.009	−0.72
	CI_95%_	0.024	0.027	0.010	0.97	0.029	0.030	0.004	0.43	0.028	0.033	0.013	0.99
MBPM	mean	1.041	1.030	−0.005	−0.51	0.897	0.903	0.0002	0.01	1.308	1.304	−0.004	−0.31
	CI_95%_	0.043	0.040	0.011	0.97	0.035	0.036	0.004	0.48	0.032	0.035	0.017	1.32

## Data Availability

The data presented in this study are available on request from the corresponding author. The data are not publicly available due in accordance with consent provided by participants on the use of confidential data.

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
