# Peer review of "Effect of 24-Week, Late-Evening Ingestion of a Calcium-Fortified, Milk-Based Protein Matrix on Biomarkers of Bone Metabolism and Site-Specific Bone Mineral Density in Postmenopausal Women with Osteopenia"

_nutrients, 2022, doi:10.3390/nu14173486_

Round 1
Reviewer 1 Report
This research evaluates the benefit of the bedtime ingestion of a calcium-fortified, milk-based protein matrix on site-specific BMD in postmenopausal women with osteopenia.
This article is based on a previous one where the authors evaluated the acute (4h) effect of bedtime ingestion of the same product on the diurnal pattern of biomarkers of bone resorption (CTX) and formation (P1NP). In this second study, they broader the results by studying the effect on bone mineral density after 24 weeks of intervention.
The authors did not mention other similar studies developed and their results with similar products controlling the consumption time. Then, is not clear the novelty of this study.
The study was correctly designed, and the analyses were performed with the correct technical approach. Nevertheless, the author could improve the discussion section by justifying possible causes of the lack of results on the main objective of the study.
Specific comments
Introduction:
- Please clarify if currently exists other similar studies, a part of the previous developed for the same authors, with similar food and control time consumption in postmenopausal women.
Materials and Methods
- Do you collect information about the months that pass by menopause? It could be interesting to include this information in Table 1 and evaluate possible differences between groups.
- Did you collect information about the physical activity developed during the intervention? It is an important variable which should be under control during the intervention. If you have this data, please add it to table 1.
- Check the p-value at the bottom of Figures 2 and 3. It is p<0,05.
Results
- Subject Characteristics: Please clarify if you identify significant differences in the basal situation between groups. These possible differences could explain the lack of differences between groups after the intervention.
Discussion
This part of the article should be improved.
- The paragraph: “to assess whether the subject.…” This information should be included in the results section, not in the discussion section. Please also avoid repetition.
- The authors should try to explain the possible causes of the results obtained and the lack of changes in BMD.
- Please compare the results with similar interventions (just in case they exist).
- Highlight the strengths and weaknesses of this study and propose improvements for future similar studies.
Author Response
Comments and Suggestions for Authors
This research evaluates the benefit of the bedtime ingestion of a calcium-fortified, milk-based protein matrix on site-specific BMD in postmenopausal women with osteopenia.
This article is based on a previous one where the authors evaluated the acute (4h) effect of bedtime ingestion of the same product on the diurnal pattern of biomarkers of bone resorption (CTX) and formation (P1NP). In this second study, they broader the results by studying the effect on bone mineral density after 24 weeks of intervention.
The authors did not mention other similar studies developed and their results with similar products controlling the consumption time. Then, is not clear the novelty of this study.
The study was correctly designed, and the analyses were performed with the correct technical approach. Nevertheless, the author could improve the discussion section by justifying possible causes of the lack of results on the main objective of the study.
Response: The reviewer’s comments on this m/s are welcomed by the research group.
Specific comments
Introduction:
- Please clarify if currently exists other similar studies, a part of the previous developed for the same authors, with similar food and control time consumption in postmenopausal women.
Response: If we read this correctly, we have not conducted studies using similar food and control time consumption with a primary objective to improve bone health in postmenopausal women. We have conducted studies using similar food, for a similar duration of intervention, but the feeding times were different and optimised for muscle protein synthesis (MPS) as the primary outcome in protection of sarcopenia in postmenopausal women (see doi:10.3945/jn.115.219022) and for which we noted a secondary outcome with respect to bone health (unpublished). Similar to the MPS study, the novelty resides in optimising the timing of the bioactive nutrient intake to the diurnal rate of bone remodelling (see further comment related to the discussion)
Materials and Methods
- Do you collect information about the months that pass by menopause? It could be interesting to include this information in Table 1 and evaluate possible differences between groups.
Response: Thank you for this suggestion. These data were included in the preliminary clinical screening of participants in the preceding study but, unfortunately, not continued for this study. The group size is relatively small to conduct such an analysis, so I think it unlikely that we would find an association
- Did you collect information about the physical activity developed during the intervention? It is an important variable which should be under control during the intervention. If you have this data, please add it to table 1.
Response: A further interesting comment relating to the 2 main modifiable, environmental factors influencing bone health, i.e., diet and PA. These data were included in the preliminary screening of subjects prior to intervention (by on-body SensewearÔ Technology) but were not tracking during the intervention.
- Check the p-value at the bottom of Figures 2 and 3. It is p<0.05
Response: Thank you, corrected in the revised m/s.
Results
- Subject Characteristics: Please clarify if you identify significant differences in the basal situation between groups. These possible differences could explain the lack of differences between groups after the intervention.
Response: Thank you, this now included in the revised m/s – last line of 1st paragraph in section 3.1
Discussion
This part of the article should be improved.
- The paragraph: “to assess whether the subject.…” This information should be included in the results section, not in the discussion section. Please also avoid repetition.
- The authors should try to explain the possible causes of the results obtained and the lack of changes in BMD.
- Please compare the results with similar interventions (just in case they exist).
- Highlight the strengths and weaknesses of this study and propose improvements for future similar studies.
Response: Thank you for these suggestions. We embrace your comments by addition of the following to the discussion, refer to a consensus paper on the advances in chronotherapy, recently published in JMBR (ref 30) and conclude with a recommendation for future studies to consider a similar approach.
The present RCT found no significant change in site-specific bone mineral density or trabecular bone score resulting from the 24-week period of supplementation with MBPM. This outcome is consistent with other RCTs of supplemental calcium that report no effect of increased calcium intake on the rate of bone loss [29]. Indeed, there is considerable debate as to whether the typical dietary range of calcium intake, or supplemental calcium intake, influences the preservation or therapeutic change in bone mineral density in postmenopausal women. Following menopause, it is the reduction in estrogen that is the primary cause of loss of bone mass, resulting in an inability of osteoblasts to maintain rates of bone formation equivalent to those of resorption [30]. Supplemental calcium intake may assist in buffering the increased calcium loss, but the deposition of calcium would be dependent on the restoration, and possible increase, in osteoblastic activity. In this respect, the high bone remodeling rate that is characteristic of postmenopausal osteoporosis, uncoupled to favor net resorption, exacerbates the rate of bone loss. A reduction in homeostatic remodeling rate, as observed in this study, may serve to reduce the rate of bone loss but, in the absence of a stimulus to osteoblastic activity may not be sufficient to restore bone mass. Supplemental calcium may, therefore, be surplus to requirement for bone tissue, leading to calcium excretion, or deposition in extraskeletal tissue. The extant evidence from long-term RCTs of calcium supplementation in support this view [5,6]. These studies report only a modest increase in BMD that is independent of habitual calcium intake [28], evident primarily during the first 12 months [31], do not progress over time, and can be explained by a reduction in the bone remodeling space and skeletal space for calcium deposition [8].
Bone metabolism exhibits potent circadian rhythms, modifiable by dietary intake. Notable in the outcome of our previous [13] study is the strong bidirectional and temporal relationship between calcium levels and serum PTH that contribute to the rhythmic activity in bone resorption. We considered the non-habitual addition of bed-time calcium and suppression of the peak night time levels of PTH which, if repeated chronically over time (24 w), would result in downregulation of bone remodeling rate, as demonstrated by the lower post-intervention serum CTX. A chronotherapy approach is supported by recent studies of the acute effect of timing of drug administration (the recombinant PTH, teriparatide™) on the diurnal rhythm of biomarkers of bone remodeling [32]. In a similar approach to that adopted in the current study, a follow-up, 12-month, intervention optimizing the timing of drug intake was found to affect a greater increase in bone mineral density in postmenopausal, osteoporotic women [33]. On the basis of these studies, we advocate the coordination of the timing of dietary intervention to the diurnal rate of bone remodeling as an effective strategy to improve the therapeutic efficacy of nutrient interventions in postmenopausal women with osteopenia.
- Conclusions
A 24-week RCT investigating the long-term effects daily, bed-time ingestion of a milk-based, protein supplement fortified with calcium and vitamin D, in postmenopausal women with osteopenia resulted in a statistically significant reduction in biomarkers of bone remodeling, but no change in site-specific bone mineral density or trabecular bone score. Of note, the magnitude of this effect approximates to 50% of that achieved by pharmacological intervention over a similar time period. We conclude that a late-evening supplement of calcium-fortified milk protein effects a beneficial decrease in the homeostatic rate of bone remodeling in persons at risk of degenerative bone disease and recommend that future experimental protocols, investigating the effect of nutrient supplementation on bone metabolism and bone health, should consider a chronotherapeutic approach to their design and implementation.

Reviewer 2 Report
The manuscript is well-written and the data presented very clearly. The study was well done and the conclusions are supported by the results. I, do, however, find issues with subject selection and with data and analysis not cited for measurements presented. All of my comments to the authors are below:
2.2: and Fig. 1: Explain how ULBC subjects from a previous study could be contacted for participation in this study? Indicate whether the informed consent from the previous study included the possibility of re-contact for future studies?
Also, how were the subjects randomly assigned to groups?
2.3: second paragraph and elsewhere: It is my opinion that “sachets” is not an apt description. The word “packets” should be used.
2.4: Please explain the use of a “phantom block” for calibration.
3.1: l.4. What range of BMD for L1-L4 is considered confirmatory of osteopenia? Also provide a reference for that range.
3.2: I note that vitamin D intake is extremely low for many subjects in the study. To what extent does that influence the study? Is the study indeed accurately titled or would a better title include “upon subjects with a habitually low intake of vitamin D”?
Last paragraph on p.8: the wording in l.4 does not make sense to me. I think it should be re-composed..
Page 9, l. 2-4: The discussion of vitamin D blood levels occurs without the presentation of data. The data should either be included in the Methods and Results or omitted.
Also, the blood levels cited are difficult to reconcile with the dietary intake of the subjects. I would expect the levels to be much lower.
Author Response
The manuscript is well-written and the data presented very clearly. The study was well done and the conclusions are supported by the results. I, do, however, find issues with subject selection and with data and analysis not cited for measurements presented. All of my comments to the authors are below:
Thank you for the positive review and overall summary of our work.
2.2: and Fig. 1: Explain how ULBC subjects from a previous study could be contacted for participation in this study? Indicate whether the informed consent from the previous study included the possibility of re-contact for future studies?
Response: The ULBC database is fully GDPR compliant. Participants’ written consent is obtained to allow contact from the researchers regarding future studies.
Also, how were the subjects randomly assigned to groups?
Response: As per 2.1., participants were block randomised (alternate blocks of 4 participants to each group). For clarity, and to indicate the temporal sequence, this has been re-stated in 2.2.
2.3: second paragraph and elsewhere: It is my opinion that “sachets” is not an apt description. The word “packets” should be used.
Response: “Sachets” was carried over from the previous study and is a product of the UK/US usage, i.e., sachet definition: UK 1. a small closed container made of paper or plastic, containing a small amount of something, but we would be willing to change to “packets” (UK sachet) US a small closed container made of paper or plastic, containing a small amount of something, usually enough for only one occasion, and have modified the m/s throughout.
2.4: Please explain the use of a “phantom block” for calibration.
Response: This should read “phantom spine” (corrected in the revised m/s). A phantom spine is a graded density bar representing the AP segment of a human spine, used in the calibration and quality assessment of a DXA scanner. Each phantom spine has a specific code issued by the manufacturer for the purpose of intra- and inter-scanner comparison. As this is an ISCD standard procedure for clinical densitometry, to which reference is made in this, and previous, m/s, further detail has not previously, and would not normally, be required.
3.1: l.4. What range of BMD for L1-L4 is considered confirmatory of osteopenia? Also provide a reference for that range.
Response: As per ISCD guidelines, we employ the manufacturer’s female normative database and use their databases for the lumbar spine as the reference standard for the resultant T-scores, define the participants age-related bone health. These are comprehensive age-related reference tables and differ between manufacturers (as to the BMD score), so the preference is to adopt the T-score rating, not the actual BMD score. At the level of the individual, we consider the change in BMD, or bone remodelling rate, to be the primary measures of bone health, not the change in T-score, i.e. increasing a T-score from 1.01 to 0.99 following intervention may re-classify a subject’s bone health, but the person remains with low bone mineral density. In this respect, the latest ISCD recommendation suggests the term “osteopenia” is retained for T-scores <1.0 to >2.5, but “low bone mass” or “low bone density” is preferred. Thus, your comment, raised in discussion, was very insightful and we did consider modifying the title and text to ‘women with low bone density’ but this would be at odds with the preceding paper, and may have led to confusion.
3.2: I note that vitamin D intake is extremely low for many subjects in the study. To what extent does that influence the study? Is the study indeed accurately titled or would a better title include “upon subjects with a habitually low intake of vitamin D”?
Response: In our opinion, the VitD status, i.e. serum 25(OH), is the important determinant of Vit D adequacy. Also see response below
Reported Vit D intakes are similar to those of other nationally representative studies doi.org/10.1079/PHN2005837 (cited as ref 16) and the range is high between subjects. Supplementation was the major contributor to overall vitamin D intake. Those with higher reported Vit D intakes report use of Vit D supplements. This may explain in part the range of intakes. The prevalence of multivitamin / single nutrient use is high in this population generally. Supplementation was the major contributor to overall vitamin D intake. Among women not consuming a vitamin D-containing supplement, mean daily intakes were approximately half the current recommendation. This aligns with other Irish national dietary research DOI: 10.1079/PHN2005837.
We have inserted the following into the discussion to address this issue.
Reported Vit D intakes were similar to those of other nationally representative studies [16]. Supplementation with Vit D was the major contributor to overall vitamin D intake. The prevalence of multivitamin / single nutrient use is high in this population generally. Among women not consuming a vitamin D-containing supplement, mean daily intakes were approximately half the current recommendation and aligns Irish national dietary data [16].
Last paragraph on p.8: the wording in l.4 does not make sense to me. I think it should be re-composed..
Response: We agree and have modified accordingly in the revised m/s by removing the qualifying no-sense phrase.
Page 9, l. 2-4: The discussion of vitamin D blood levels occurs without the presentation of data. The data should either be included in the Methods and Results or omitted.
Response: Baseline serum 25(OH)VitD are presented in section 3.3. (copied below) indicating the range across all subjects, mean values for CON and MBPS and status with respect to the 50nmol/L reference. The method of analysis of 25(OH)VitD is described in section 2.5 (copied below). As indicated in the discussion the 25(OH)VitD status was ascertained at baseline to indicate participants’ ability to effectively absorb dietary calcium only.
Serum 25(OH)vitamin D ranged from 40.0 to 113.0 nmol/L. Only 3 of the 65 participants recorded a 25(OH) vitamin D below 50nmol/L. No difference in 25(OH) was observed between CON (72.9 ± 16.0) and MBPS (70.3 ± 14.8) at baseline (P = 0.490).
Serum CTX, P1NP and total 25-hydroxyvitamin D was analysed by 2-site immunometric assay using electrochemiluminescent detection (Roche Cobas e411, Roche Diagnostics, UK. The interassay CV was 5.3% for CTX, 4.5% for P1NP and <8% for 25(OH) Vit D.
Also, the blood levels cited are difficult to reconcile with the dietary intake of the subjects. I would expect the levels to be much lower.
Response: We would, in part, concur, but are mindful of the relatively short period of dietary recall used in this study to assess habitual dietary intake of Vit D. On reflection, an FFQ might have been a better option. Millen and Bodnar (doi.org/10.1093/ajcn/87.4.1102S) conclude in their review of vitamin D assessment in population-based studies that dietary assessment must cover intake over a long enough period of time (eg, at least a 3-mo) to capture fully less commonly consumed foods (i.e., fatty fish) and classify persons with respect to their usual vitamin D intake. Change in serum 25(OH)Vit D in response to habitual dietary intake of VitD is a relatively slow process and we consider the serum 25(OH) Vit D levels to better reflect the VitD status of the subjects on entry to the study (also see comment above).